# A 2022 Systematic Review and Meta-Analysis of Enriched Therapeutic Diets and Nutraceuticals in Canine and Feline Osteoarthritis

**DOI:** 10.3390/ijms231810384

**Published:** 2022-09-08

**Authors:** Maude Barbeau-Grégoire, Colombe Otis, Antoine Cournoyer, Maxim Moreau, Bertrand Lussier, Eric Troncy

**Affiliations:** 1Groupe de Recherche en Pharmacologie Animale du Québec (GREPAQ), Département de Biomédecine Vétérinaire, Faculté de Médecine Vétérinaire, Université de Montréal, Saint-Hyacinthe, QC J2S 7C6, Canada; 2Unité de Recherche en Arthrose, Centre de Recherche du Centre Hospitalier Universitaire de L’Université de Montréal, Montréal, QC H2X 0A9, Canada

**Keywords:** osteoarthritis, nutraceuticals, enriched diets, pain, animal, methodological quality, scientific evidence, metrological validation

## Abstract

With osteoarthritis being the most common degenerative disease in pet animals, a very broad panel of natural health products is available on the market for its management. The aim of this systematic review and meta-analysis, registered on PROSPERO (CRD42021279368), was to test for the evidence of clinical analgesia efficacy of fortified foods and nutraceuticals administered in dogs and cats affected by osteoarthritis. In four electronic bibliographic databases, 1578 publications were retrieved plus 20 additional publications from internal sources. Fifty-seven articles were included, comprising 72 trials divided into nine different categories of natural health compound. The efficacy assessment, associated to the level of quality of each trial, presented an evident clinical analgesic efficacy for omega-3-enriched diets, omega-3 supplements and cannabidiol (to a lesser degree). Our analyses showed a weak efficacy of collagen and a very marked non-effect of chondroitin-glucosamine nutraceuticals, which leads us to recommend that the latter products should no longer be recommended for pain management in canine and feline osteoarthritis.

## 1. Introduction

Osteoarthritis (OA) is a widespread musculoskeletal disorder in pets [1]. In the absence of a curative treatment, veterinarians attempt to control the symptoms of pain. Therapeutic goals therefore focus on reducing joint pain and improving motor function to increase the quality of life of the affected animals. The most recommended drugs are non-steroidal anti-inflammatory drugs (NSAIDs) based on their efficacy [2,3]. However, compliance with this treatment is difficult due to repeated (often daily) administration; side effects are not uncommon (mainly gastrointestinal irritation, nephrotoxicity, hepatotoxicity); and the benefits of long-term management on the longevity and quality of life remain limited [4]. 

To the best of the authors’ knowledge, no therapeutic approach has any indication of a delayed effect on the progression of OA. Thus, the terms “chondroprotective”, “structure-modulator” or “disease-modifying” do not yet apply to the therapeutic approaches available in pet OA, with all therapeutic indications revolving around an improvement in the behavioural or physiologic signs associated with OA pain.

The lack of alternatives in OA therapy would benefit from an evidence-based statement on the different approaches available and their potential benefits for OA-affected animals. Fortified diets and nutraceuticals have gained popularity among the veterinary community in recent decades. Indeed, the field of nutraceuticals has experienced a rapid and substantial economic growth. The increase in the consumption of natural substances is mainly associated with the rise of owners’ awareness of their health and lifestyle beliefs, which they transpose to the care of their animals as well [5,6]. The global veterinary dietary supplements market was valued at USD 1.6 billion in 2020 and is expected to continue growing with an estimated annual growth rate of 8.2% by 2028 [7]. Within the veterinary recommendations of nutraceuticals use, OA and degenerative joint disorder are the diseases for which veterinary practitioners most commonly emit a recommendation [8,9]. However, regulatory assessments of these compounds primarily focused on the absence of side effects (safety), quality and nutrition but did not require proof in therapeutic efficacy [10,11]. 

This review focused on fortified therapeutic diets as well as nutraceuticals, i.e., products made from food substances, available in a wide variety of formulations such as tablets, capsules, drops, powders, treats or other medicinal forms usually not associated with food, which have been shown to have a possible beneficial or protective pharmacological effect against chronic diseases.

Three previous systematic reviews on the treatment of animal OA revealed a disappointing quantity and quality of scientific evidence regarding fortified therapeutic diets and nutraceuticals [12,13,14]. The evidence from these three systematic reviews on the use of these products was not strong enough to adopt or support meaningful recommendations.

The aim of this systematic review and meta-analysis was to examine the evidence for analgesic efficacy of fortified therapeutic diets and nutraceuticals to build up solid research evidence (evidence-based medicine) and to properly disseminate findings on the efficacy of the therapeutic potential in dogs and cats affected by OA. This project, therefore, benefits from the addition of an objective and quantitative assessment of quality and efficacy, which allows conclusions to be drawn that are supported by good scientific evidence. Our hypothesis was that, in 2022, we have sufficient evidence to support, or not, the use of therapeutic diets or nutraceuticals in the management of canine and feline OA.

## 2. Methods

### 2.1. Literature Search and Inclusion of Studies

Four electronic databases (CAB-Abstract, Embase, Global-Health and Pubmed) were searched for articles published from 1980 to 10 October 2021. A systematic search was performed using the following predefined terms: (arthrosis OR osteoarthr* OR “degenerative joint disease”) AND (cat OR cats OR feline) AND (dog OR dogs OR canine) AND (“disease modifying agent” OR nutrient* OR nutritional OR “nutritional medicinal product” OR “nutritional supplements” OR nutraceutical* OR “botanical drugs” OR “botanical food supplements” OR “herbal health nutritionals” OR “herbal health nutritional” OR “herbal medicine” OR “fortified food” OR “food additive” OR “food additives” OR diet OR “dietary supplements” OR “dietary supplement” OR dietary OR “geriatric diet” OR “natural product” OR “natural products” OR phytotherapy OR “complementary medicines” OR “complementary medicine” OR homeopathy OR antioxidant OR “food derived products” OR “food derived product” OR “mineral supplements” OR “mineral supplement” OR supplement OR supplements). All duplicates present between the different databases were removed in the selection of articles (Figure 1). A few articles from complementary internal sources were also added (*N* = 20). The references were then all transferred to the EndNoteTM X9 platform (Clarivate Analytics, Philadelphia, PA, USA). 

All articles were then assessed for inclusion. Briefly, the selected studies had to test the effect of nutraceuticals, or therapeutic diets, on canine or feline OA pain. Induced OA models were also included. If a research article had multiple treatment arms (i.e., multiple compounds or doses under study), each trial was assessed and independently analysed. The data from the included studies were extracted using a standardised format for the assessment of trial quality and analgesic efficacy. Two reviewers (M.B.-G., A.C.) independently extracted the data; discrepancies were identified and resolved by discussion with a third reviewer (C.O.), if necessary.

### 2.2. Selection of Articles

Figure 1 shows the PRISMA flowchart of the identified studies. 

While 57 articles on canine and feline OA were selected, a total of 72 trials, due to the different arms tested in many studies, and 38 different compounds were evaluated. Whereas 69 of these trials used a canine model, only 3 were tested in cats.

### 2.3. Meta-Analysis: Construction and Validation of Analysis Scales for Data Extracted from Publications

A “quality of the trial” meta-analysis scale was developed, based on three evaluation criteria, in order to assess each therapeutic trial in a systematic, independent and quantitative manner. A meta-analysis scale “analgesic efficacy” was also constructed in the form of a simple categorisation (see below) of the effect of the treated group vs. control group, temporal (within-group) improvement and non-effect. The assessment grids were developed based on the models used in three previous systematic reviews [12,13,14], in compliance with ARRIVE recommendations (Animal Research: Reporting In Vivo Experiments; [15]), CONSORT (Consolidated Standards of Reporting Trials; [16]) and CAMARADES (Collaborative Approach to Meta Analysis and Review of Animal Data from Experimental Studies; [17,18]). Finally, the study was registered on the international prospective register of systematic reviews PROSPERO (www.crd.york.ac.uk/prospero/; accessed on 4 February 2022, CRD42021279368) whose educational tools guide the systematic review process. The report of the information collected followed the PRISMA guide (Preferred Reporting Items for Systematic reviews and Meta-Analyses; [19,20]).

After the primary construction of the quality of trial scale, it was subjected to a series of validation (face validation, internal and external content validation) by several independent evaluators, as well as construct validation. Once the development of the grids was fully completed (Table 1), all the articles selected (with the different product and dose trials) were evaluated and scored by three independent investigators considered to have different levels of expertise (M.B.-G., C.O. and E.T.). The values assigned by each evaluator were compared for each trial tested, and the single final score, used for the statistical analyses, was systematically obtained by consensus.

### 2.4. Quality of Trials Assessment Scale (Table 1)

The assessment grid consisted of three sections, seeking to test three fundamental criteria: risk of bias, methodological quality and strength of scientific evidence. The quality total score was obtained by adding the scores of the three constituent criteria, and all trials were classified into four quality levels based on the distribution of the totals obtained.

### 2.5. Efficacy Assessment Scale

The evidence of efficacy, or non-effect, of the compound tested was supplemented by a simple categorisation: (1) trials with “analgesic effect” represented an improvement in the condition of the animal with the treatment, over time and compared with a control group. This is, therefore, an inter-group temporal comparison. (2) Trials with “improvement” represented within-group improvement in condition over time. Animals are, therefore, only temporally evaluated. Considering that chronic conditions, such as OA, are subject to changes over time, this effectiveness was less than the previous one. (3) Finally, the trials with a “non-effect” did not represent any improvement, neither over time within the same group nor between the groups. This assessment was also systematically carried out by consensus of the three assessors.

### 2.6. Statistical Analyses

All trials were grouped into nine categories as shown in Table 2: 1. omega-3-enriched therapeutic diets (*N* = 10); 2. omega-3-based nutraceuticals (fish oil, green mussels, etc.) (*N* = 10); 3. collagen-based nutraceuticals (*N* = 11); 4. nutraceuticals based on chondroitin–glucosamine (*N* = 9); 5. cannabinoid-based nutraceuticals (*N* = 7); 6. nutraceuticals based on hydroxycitric acid (*N* = 3); 7. nutraceuticals based on calcium fructoborate (*N* = 3); 8. composite nutraceuticals (*N* = 3); and 9. others (*N* = 16). Only categories (ctg.) 1 to 5 were kept for comparison of quality and efficacy, as the others (ctg. 6 to 9) did not present a sufficiently large number of trials (*N* ≤ 3).

**Table 2 ijms-23-10384-t002:** Presentation, by category, of clinical trials on therapeutic nutrition and nutraceuticals in canine and feline osteoarthritis.

Categories and Compounds Tested	References
Category 1. Omega-3-enriched therapeutic diets	
Green-lipped mussels	[21,22,23,24]
Fish oil	[25,26,27,28,29]
Category 2. Omega-3-based nutraceuticals	
Green-lipped mussels	[21,30,31,32,33,34]
Fish oil	[35,36,37]
Category 3. Collagen-based nutraceuticals	
Collagen	[38,39,40,41]
Collagen, glucosamine hydrochloride and chondroitin sulphate	[40,42]
Collagen-derived gelatine	[43]
NEM^®^	[44]
Ovopet^®^	[45]
Movoflex^TM^	[46]
Category 4. Nutraceuticals with chondroitin-glucosamine	
Chondroitin sulphate	[30]
Glucosamine hydrochloride, chondroitin sulphate and manganese	[47]
Glucosamine hydrochloride and chondroitin sulphate	[40,42,48]
Glucosamine hydrochloride, chondroitin sulphate, N-acetyl-D-glucosamine, ascorbic acid and zinc sulphate	[49,50]
Glucosamine hydrochloride, chondroitin sulphate and hyaluronic acid	[51]
Glucosamine hydrochloride, chondroitin sulphate and avocado and soya unsaponifiables	[52]
Category 5. Cannabinoid-based nutraceuticals	
Cannabidiol	[53,54,55,56,57]
Category 6. Nutraceuticals based on hydroxycitric acid	
Hydroxycitric acid	[39]
Hydroxycitric acid and chromemate	[39]
Hydroxycitric acid, chromemate and collagen	[39]
Category 7. Nutraceuticals based on calcium fructoborate	
Calcium fructoborate	[58]
Calcium fructoborate, glucosamine hydrochloride and chondroitin sulphate	[58]
Category 8. Composite Nutraceuticals	
Flexodol^®^/Flexxil^®^	[59]
Dinamic^TM^	[60]
Curcuvet^®^-boswellic acid-glucosamine-chondroitin-omega-3-Vit. C, E-*Saccharomyces cerevisiae*	[61]
Category 9. Others	
Special protein milk concentrate	[62]
Curcumoids	[63]
Elk velvet antler	[64]
*Boswellia serrata* extracts	[65]
Avocado and soybean unsaponifiables	[66]
Yeast (Beta-1.3/1.6 glucans)	[67]
*Brachystemma calycinum* D don extracts	[68,69]
STA-LITE^®^ polydextrose	[70]
S-adenosyl L-methionine (SAMe)	[71]
Crominex 3+ ^®^ (chrome trivalent, *Phyllanthus emblica*, shilajit)	[72]
Shilajit (*Asphaltum punjabianum*)	[73]
Vitamin E	[74]
*Terminalia chebula* (Indian myrobolan)	[75]
Diets enriched with curcumoid extract, hydrolysed collagen and green tea extract	[76]
4CYTE^TM^ Epiitalis^®^ Forte (*Biota orientalis*)	[77]

The null hypothesis was that no statistically significant difference existed between the scores of the five categories for trial quality or analgesic efficacy. For statistical analyses, we used R^®^ software (Version 4.0.3, R Foundation for Statistical Computing, Vienna, Austria) with an alpha threshold of 0.05 for the significance of the results.

#### 2.6.1. Quality of Trials

In the process of the construct validation of the grid, we tested the correlation links between the three constituent criteria as well as the links between these criteria and the quality total using linear mixed models (LMMs), integrating the trial identifier as a random factor to control for pseudo-replication bias. Finally, an LMM with the trial identifier as a random factor tested the effect of the category (1 to 5) on the quality total. Tukey’s post hoc tests without correction for multiple comparisons were then performed, in an exploratory manner, to identify the pairs that were significantly different between categories. These analyses only considered publications with canine subjects as we did not want to combine species in the analyses, and dogs represented most of the trials listed.

#### 2.6.2. Analgesic Efficacy

The descriptive analyses initially focused on all the trials included in categories 1 to 5, without considering their quality, indicating the level of efficacy in the percentage of effect, improvement and non- effect. A weighing method (Table 3) was then applied to give more weight to the efficacy results obtained on the better-quality trials.

Generalised linear models (GLMs) tested the effect of each category (1 to 5) on efficiency in the interaction with quality total, again only using the canine publications. Here, the dependent variable efficacy was considered in two different ways: first, when there was an effect only and, second, when there was an effect or improvement. With the dependent variable being, in both cases, binomial, we used a logit link in the GLMs. A mixed proportional odds (POM) model finally identified the differences between categories on effectiveness, if any.

#### 2.6.3. Complementary Analyses

The effects of follow-up duration, dose used in each trial and quality total on efficacy were analysed for the dog trials of categories 1 to 5 as the dosage and duration varied according to the species, and most of the trials focused on dogs. A POM model was again used. The effect size was also calculated for these same 5 categories, this time including all trials, for the efficacy data using SPSS software (Version 27.0; IBM Corp. SPSS Statistics for Windows, Armonk, NY, USA). The measure chosen was Cohen’s d with the global variance as the normaliser. First, the efficacy scores of the different categories were compared with the scores of the negative controls of these same trials, which were then scored as inversely described above for the test article.

Secondly, the 5 product categories were also compared with each other, again based on their effectiveness score. The interpretation of the results was made because of the benchmarks suggested by Cohen [78].

## 3. Results

### 3.1. Validation of the “Quality of Trial” Scale

The statistical validation of the scale was carried out considering only the publications in the canine OA, but the publications on cats revealed the same tendencies. The criteria “methodological quality” and “strength of scientific evidence” were significantly associated (LMM: F = 13.29; CI^95%^ = [0.15; 0.49]). Neither of the other two relationships tested between the constituent criteria was significant. In addition, all three constituent criteria of the scale had a positive and significant link with the “quality total”, with the most associated criterion (R^2^ = 0.81) being “methodological quality”. This indicated that each criterion was significantly involved in the composition of the quality of trial scale and justified the use of the quality total as a variable reflecting the quality of each trial. 

### 3.2. Quality Assessment

#### 3.2.1. Descriptive Distribution of Quality

Following the classification of the quality total into four levels, the number of trials per quality level was balanced. Thus, there were 38 high-quality trials (grouping levels A “very high” and B “good”) and 34 trials of mediocre quality (levels C “medium” and D “low quality”). 

For compound categories 1 to 5 (see Table 2 for details), similar proportions were also observed: 28 high-quality trials (levels A and B) and 19 mediocre-quality trials (levels C and D). However, this distribution was not homogeneous between the five categories as shown in Figure 2.

Collagen-based (ctg. 3) and chondroitin-glucosamine-based (ctg. 4) nutraceuticals stand out with a higher presence of lower-quality trials. The quality level ratios (the number of AB/CD level trials) were 0.6 and 0.8, respectively, for these two categories. Conversely, omega-3-enriched therapeutic diets (ctg. 1), omega-3-based nutraceuticals (ctg. 2) and cannabinoids (ctg. 5) had more high-quality trials as evidenced by the quality level ratios of 4.0, 2.3 and 2.5, respectively. 

For the remaining categories, the quality level of the trials varied between them. Hydroxycitric acid nutraceuticals (ctg. 6) had three low-quality trials (level D). Calcium fructoborate nutraceuticals (ctg. 7) had three very high-quality trials (level A). Composite nutraceuticals (ctg. 8) included one very high-quality trial (level A) [59] and two medium-quality trials (level C).

Finally, in the category of other products (ctg. 9), out of the 16 trials selected, there were three very high-quality (level A) [63,69,76], three good-quality (level B) [64,68,71], seven medium-quality (level C) and three low-quality (level D) trials [[67],[70],[75] ].

#### 3.2.2. Effect of the Category on the Quality Total

The descriptive differences observed in Figure 2 were confirmed by statistically significant differences between the trials on canine OA. Collagen-based nutraceuticals (ctg. 3) showed a lower quality total (median 18.5 [min = 13.0; max = 32.0]) than omega-3-enriched therapeutic diets (ctg. 1) (31.5 [13.5; 38.0], *p* = 0.043) and omega-3-based nutraceuticals (ctg. 2) (33.0 [14.0; 44.0], *p* = 0.026). The quality total of the other categories lied between these two extremes, namely chondroitin-glucosamine nutraceuticals (ctg. 4) (23.0 [16.0; 40.0]) and cannabinoid nutraceuticals (ctg. 5) (26.0 [18.5; 37.5]).

Regarding the three trials on feline OA, they all showed a good quality level. The trial presenting omega-3-enriched therapeutic diets (ctg. 1) was of very high quality [23], while the other two on omega-3-based (ctg. 2) and chondroitin-glucosamine nutraceuticals (ctg. 4) were of good quality [36,50].

### 3.3. Analgesic Efficacy Assessment

#### 3.3.1. Descriptive Distribution of Efficacy

Figure 3 shows the distribution of efficacy, i.e., effect (compared with a control group), improvement (within-time) and non-effect for all trials included in categories 1 to 5. It was observed that omega-3 nutraceuticals (ctg. 2) stand out in terms of effect, while chondroitin-glucosamine nutraceuticals (ctg. 4) stand out for their lack of efficacy with 88.9% non-effect and 0% effect. The other categories showed a reduced percentage of non-effect: from 10.0% for omega-3-enriched therapeutic diets (ctg. 1) and omega-3-based nutraceuticals (ctg. 2) to 18.2% for collagen-based nutraceuticals (ctg. 3) and 14.3% for cannabinoid-based nutraceuticals (ctg. 5).

For categories 6 to 8 (*N* = 3, each), as for the quality level, the efficacy was also variable. Hydroxycitric acid (ctg. 6) used alone had no effect and only showed improvement when combined with chromemate, alone or with collagen as well [39]. None of the three trials of calcium fructoborate nutraceuticals (ctg. 7) had any effect, whether used alone or in combination with chondroitin-glucosamine [58]. For composite nutraceuticals (ctg. 8), the combination of phytotherapeutic extracts with omega-3, chondroitin-glucosamine, vitamins, etc. showed an analgesic effect [59,60] and a non-effect [61].

Finally, of the 16 trials of other products (ctg. 9), four detected an analgesic effect with special protein milk concentrate [62], elk velvet antler [64], avocado and soy unsaponifiables [66] and vitamin E [74]; nine detected a simple improvement; and three detected a non-effect for turmeric [63], STA-LITE polydextrose [70] and S-adenosyl L-methionine [71] compounds.

#### 3.3.2. Effect of Category on Trial Efficacy

In both the GLMs tested on the canine trials, category had a significant effect on analgesic effectiveness, whether restricted to effect alone (goodness-of-fit = 0.248, LRT 2 = 12.74, df = 4, *p* = 0.013) or when combining effect + improvement (goodness-of-fit = 0.500, LRT 2 = 16.31, df = 4, *p* = 0.003). The results of the subsequent analysis of the quality-adjusted efficacy scores also showed that the level of efficacy of the chondroitin-glucosamine nutraceuticals (ctg. 4) was significantly lower than those of the other four categories (estimated = 3.96 (1.06), Z = 3.72, *p* < 0.001) as shown in Figure 4.

Figure 4 shows, without significance, that omega-3-based nutraceuticals (ctg. 2) had the highest level of efficacy (mean 3.3 ± 1.9), followed by omega-3-enriched therapeutic diets (ctg. 1) (2.4 ± 2.7) and cannabinoid nutraceuticals (ctg. 5) (2.0 ± 2.2). Collagen-based nutraceuticals (ctg. 3) (1.0 ± 2.2) showed the lowest efficacy, while chondroitin-glucosamine nutraceuticals (ctg. 4) (−2.7 ± 2.0) were ineffective. 

#### 3.3.3. Complementary Analyses

Complementary analyses were conducted only on canine articles as the duration and dosage could not be compared between species, and the sample on canine OA was much larger. No significant effect of treatment duration, dose and quality total of the trials was noted on efficacy in any of the five categories tested. The follow-up duration greatly varied for all categories, with the range varying between 28 and 180 days. Overall, the treatment dose had no significant effect on the level of efficacy, with doses being fairly consistent within each category. It should be noted, however, that for cannabinoid-based nutraceuticals (ctg. 5), one trial used a much lower dose (0.5 mg/kg/d), and this was the only one to show a non-effect [56].

Effect size calculations show a medium to large effect of enriched therapeutic diets (d = 0.58) (ctg. 1) and omega-3-based nutraceuticals (d = 1.19) (ctg. 2) compared with the score of controls in these same categories. A large effect was also observed, this time favouring the efficacy score of the negative control, for the collagen-based nutraceuticals (d = −1.57) (ctg. 3) and chondroitin-glucosamine-based nutraceuticals (d = −1.39) (ctg. 4) categories. Finally, no effect could be noted for cannabinoid-based products (d = 0) (ctg. 5).

The comparison between categories revealed a large effect of all categories (ctg. 1: d = 2.13; ctg. 2: d = 3.03; ctg. 3: d = 1.74; ctg. 5: d = 2.25) compared with the chondroitin-glucosamine nutraceuticals category (ctg. 4). Collagen-based nutraceuticals (ctg. 3) also appeared to have a smaller effect than omega-3-enriched therapeutic diets (d = 0.58) (ctg. 1), omega-3-based nutraceuticals (d = 1.08) (ctg. 2) and cannabinoid-based nutraceuticals (d = 0.46) (ctg. 5).

## 4. Discussion

### 4.1. Review of the Work

This systematic review and meta-analysis assessed the efficacy of 38 compounds in the treatment of clinical signs of OA. A total of 57 articles, comprising 72 trials, were analysed. 

Of the 57 articles identified, we obtained 54 articles on canine OA, while only 3 articles on the use of nutraceuticals in the context of feline OA could be found by our searches. This lack of literature on cats can perhaps be explained by the challenge in their pain evaluation. Cats are known to be less expressive, and since the domestication of cats has historically been very different from dogs, humans tend to have poorer skills in recognising painful behaviour [79]. This phenomenon is very unfortunate since these animals are as affected by OA as dogs, even though their condition is, by far, much less studied.

### 4.2. Evaluation Scales: Trial Quality and Analgesic Efficacy

Both scales were established following the review of methodologies presented in previous articles, including three systematic reviews assessing the benefits of enriched therapeutic diets and nutraceuticals in canine OA [12,13,14] (Materials and Methods). They were subsequently successful for face, content and construct validity. The latter justified the single use of the quality total in our statistical analyses. This validation assures that our meta-analysis results from systematic, independent and quantitative measures in quality and efficacy [80].

In contrast to our work, the assessment methods of the three previous systematic reviews were rather based on qualitative scales; only the review by Vanderweerd et al. [14] added an assessment of quality attributed in percentages. The enrichment of the quality of trials scale, therefore, was mainly the addition of this quantitative aspect that was missing in the previous works and follows the present rules of evidence-based medicine. In addition, several sub-criteria were developed and detailed in the construction of our evaluation scales to extract a maximum of information from each trial. This evaluation also included a test of intra-observer repeatability and inter-observer reproducibility.

The analgesic efficacy scale was simpler to establish as it had only three levels: statistically significant effect of the treatment tested vs. a control group, improvement only of the treated group over time and non-effect. It should be noted that this assessment of analgesic efficacy was based on the methods, results and statistical analyses used in each trial. Following the PROSPERO procedures helps to standardise the collection of data in the systematic review and therefore its quality. Finally, all trials were assessed and scored in a consensual manner by three observers with different levels of expertise for both grids.

### 4.3. Combination of Quality of Trials and Analgesic Efficacy (Ctg. 1–5)

Regarding the results of our analyses on categories 1–5 (*N* > 3 trials), the assessment of quality tended to be significantly impacted by the product category, while efficacy was significantly influenced by the category.

Combining the results of the quality (Figure 2) and efficacy (Figure 4) assessments allowed us to support the efficacy, associated to the quality of trials, of omega-3 nutraceuticals in supplement form (ctg. 2) or omega-3-enriched therapeutic diets (ctg. 1) and cannabinoid nutraceuticals (ctg. 5) (respectively from the most to least effective). Our analyses also showed, with studies of lesser quality, a weak efficacy of collagen-based nutraceuticals (ctg. 3) and a very marked non-effect of chondroitin-glucosamine-based products (ctg. 4). The quality of the latter studies is disappointing in terms of concluding on the use of these products, and the total lack of efficacy of chondroitin-glucosamine nutraceuticals (ctg. 4) stands out in comparison with the other categories, and therefore indicates that these products should no longer be recommended in cases of canine or feline OA.

### 4.4. Enriched Therapeutic Diets (Ctg. 1) and Nutraceuticals (Ctg. 2) Based on Omega-3

For omega-3-based compounds, i.e., categories 1 and 2, the results of all previous reviews support our inferences with a high level of comfort that these products are highly effective [12,13,14]. The one difficulty encountered by previous reviews, and the present one, is the lack of objective data available in studies of these products. However, there is a preponderance of high-quality trials within these two categories. The complexity of efficacy assessment in the studies of therapeutic diets in canine OA has already been raised [81]. Compared with other therapeutic modalities (e.g., NSAID), the rate of negative responders (using objective kinetic podobarometric assessment) to the introduction of a therapeutic diet is three times higher, while, conversely, the rate of positive responders to a placebo-control diet is up to two times higher. A therapeutic diet role is to primarily meet the dog nutritional needs while providing active ingredients susceptible to modify the condition. The contribution of a balanced diet may, therefore, have less impact on dogs recruited into a clinical trial because they are already well-nourished, while, on the other hand, the condition of dogs receiving the control diet will improve [81]. Moreover, the variability in the diet ingestion compared with that of supplements could also explain a variation in the exposure to the active ingredients, and thus in the expression of their expected benefits, not to mention inter-individual perturbations on the gut microbiota [81].

Regarding the efficacy of omega-3 products, only 2 [25,30] of the 20 trials on them were non-effective, which underlines the analgesic potential of these products. The same finding was also true concerning the feline OA trials that provided good-quality studies and both showed an analgesic effect [23,36]. Only one trial of omega-3-enriched therapeutic diets (ctg. 1) [25] showing no improvement in the OA condition of the dogs was a dose titration study, and this trial tested the lowest dose. 

This meta-analysis supports the use of omega-3 supplementation for the management of canine and feline OA. The incorporation of omega-3 into a therapeutic diet offers the ease of administration of adapted doses of omega-3, and the diets also facilitate, through their nutritional quality, the maintenance of digestive and renal functions that are often affected in these geriatric patients, while theoretically favouring excess weight loss.

### 4.5. Cannabinoid Nutraceuticals (Ctg. 5)

Trials with cannabidiol in dogs also indicated high-quality studies and good evidence of efficacy. It is interesting to note that these studies are recent (published between 2018 and 2021) and more in line with international recommendations. The efficacy of cannabidiol in the treatment of chronic pain, mainly neuropathic in nature, has already been reported in rodent models [82] and in human patients [83]. Seven trials testing cannabidiol in the management of OA pain in dogs were evaluated. All showed an improvement in the condition with the exception of a single trial conducted by Verrico et al. [56]. In this trial, the authors tested a low dose (0.5 mg/kg/day) compared with a higher dose (1.2 mg/kg/day). Interestingly, in the same study, a trial of a liposomal formulation at the same low dose (0.5 mg/kg/day) was effective. Liposomal encapsulation has already shown, in humans and mice, a better bioavailability [56]. The results of this meta-analysis are promising, but further investigation is needed to determine the efficacy, doses, formulations and combinations recommended for the treatment of canine OA pain. Further studies will also be necessary to conclude on the use of cannabinoids in cats since none have been carried out to this day.

### 4.6. Collagen-Based Nutraceuticals (Ctg. 3)

The scientific evidence for the efficacy of collagen (ctg. 3) in its UC-II (undenatured type II collagen) formulation, alone or combined most often with chondroitin-glucosamine, or in the formulation derived from an eggshell membrane [44,45,46] was the lowest of the four categories with a positive efficacy score. The main reason for this relates to the poor quality of the trials that evaluated it: small sample size (*N* = 5 dogs per group [38,39,42], *N* = 7–10 per group [40] and *N* = 9 [46]); assessment using non-validated subjective tools without observer guidance [38,39,40,42,43] with a visibly non-adapted statistical methodology; and a limited (often single) number of assessment times in the follow-up period [38,41,44,46]. In addition, one study investigated a therapeutic diet with green tea extract, turmeric and hydrolysed collagen as active ingredients, and while the subjective assessments were positive, the objective podobarometric assessment (kinetic analysis of ground reaction forces) was inconclusive [76]. It, therefore, appears impossible to rule, at the present time, on an indication for collagen in canine OA based on the results of this meta-analysis.

### 4.7. Chondroitin-Glucosamine Nutraceuticals (Ctg. 4)

The systematic review of the literature included nine trials that evaluated, mostly in combination, glucosamine hydrochloride and chondroitin sulphate. Chondroitin-glucosamine nutraceuticals (ctg. 4) showed strong evidence of non-effect and a significant statistical difference in efficacy from the other categories (ctg. 1, 2, 3 and 5) in the meta-analysis. Of the nine trials assessed in this review, only one [49] showed an improvement in the condition of the animals assessed, but this was using a non-validated subjective tool and for only one assessment time (at day 70), a difference not present before (day 14 or 42) or after (day 90). It should be noted that dosing was reduced by one third between days 42 and 70 and stopped after day 70, while the authors concluded that there was non-inferiority of the nutraceuticals vs. a positive control using carprofen in their OA dogs [49].

In the human literature, there are several criticisms of its use in OA, and a meta-analysis similarly found no effect on OA pain compared with the placebo [84]. A veterinary systematic review on the use of chondroitin-glucosamine was also inconclusive in dogs [85]. Like these previous reviews, the results of the present meta-analysis led to the conclusion that chondroitin-glucosamine nutraceuticals should not be prescribed in canine or feline OA.

### 4.8. Nutraceuticals Based on Hydroxycitric Acid (Ctg. 6), Calcium Fructoborate (Ctg. 7) and Composite Nutraceuticals (Ctg. 8)

The results of the hydroxycitric acid (ctg. 6) or fructoborate (ctg. 7) products were not conclusive. The low quality of the hydroxycitric acid trials and the lack of efficacy of the fructoborate trials did not allow one to definitively conclude on the use of these products. However, for both types of products, all trials were obtained from the same article [39,58], which could potentially bias our conclusions. Regarding composite nutraceuticals (ctg. 8), they do seem to be of interest as two of the trials showed an effect * in [59], ^ψ^ in [60]. The composition of the two composite nutraceuticals is based on a combination of herbal medicine (*Harpagophytum procumbens*, *Boswellia serrata*, *Ribes nigrum*, *Salix alba**, *Tanacetum parthenium**, *Ananas comosus**, *Lentinus edodes*^ψ^, *Equisetum arvense*^ψ^ and *Curcuma longa*), omega-3s, chondroitin-glucosamine, methylsulphonylmethane*, L-glutamine* and hyaluronic acid*. Both studies were characterised by a remarkable safety profile in *N* = 16 [59] and *N* = 10 [60] dogs treated over 2 and 3 months, respectively. The results of the Canadian study were impressive as they incorporated objective assessments (podobarometric gait analysis and actimetry), but it is regrettable that the products are not marketed [59]. In any case, more high-quality studies are needed to properly assess these products.

### 4.9. Other (Ctg. 9)

Finally, all the other nutraceuticals evaluated did not present sufficient evidence of efficacy to decide on their indication. However, some of these compounds seemed promising, with high-quality studies, such as elk velvet antler [64] or *Brachystemma calycinum* D don extracts [68,69]. The results with turmeric were conflicting, being either negative [63], ambiguous [61,76] or positive [59,60]. Although turmeric seems to benefit from a composite synergistic approach, evidence of efficacy remains to be confirmed with further studies.

### 4.10. Effect Sizes

The calculated effect sizes, in comparison with negative controls, supported the evidence of efficacy of omega-3-enriched diets (ctg. 1) and omega-3-based nutraceuticals (ctg. 2). This indicates a clinically important effect of these products.

For collagen-based nutraceuticals (ctg. 3), this comparison highlighted the uncertainty about the efficacy of these products. The calculation of the effect size took into account the level of not only efficacy but also quality for the trials. For ctg. 3, quality had a huge influence on the scores obtained. The measures collected, therefore, indicated that we cannot conclude to an effect of collagen and that further studies of high quality would be required.

The effect size obtained for the chondroitin-glucosamine nutraceuticals (ctg. 4) clearly showed the lack of efficacy of these products, with the negative controls showing even a higher averaged efficacy than the product trials. Furthermore, the comparison of the efficacy with the other categories showed a strong non-effect of these nutraceuticals. 

Finally, the results obtained for the cannabinoid-based nutraceuticals (ctg. 5) did not support a definitive conclusion on the use of these products, and further studies would, again, be necessary.

Interpretation of these effect sizes must be performed with caution as this is not a comparison of the data obtained from the evaluations of these trials but a comparison of the scores assigned. These scores tend to favour the effects, and, therefore, the evaluation of negative controls is tricky. The use of control was also not present in all the included trials, so they could not be counted in the averaged efficacy level of the categories. The lack of follow-up over time for these control groups was another constraint that was often encountered. The assessment of the efficacy level of the controls was therefore sometimes solely based on an interpretation of the results presented without the support of the statistical analyses presented in the trials.

### 4.11. Potential Mechanism of Nutraceuticals Action

Nutraceuticals’ precise mechanisms of action are still not well-determined in any target species [86,87]. Moreover, the poor application of consistency and standardisation in the nutraceuticals composition makes it difficult to conclude on the mechanisms of action underlying a single product [88,89]. Regarding OA, the favourite molecular targets focus on anti-inflammatory, anti-oxidative and anti-catabolic actions, thus sustaining the global attention to cytokine (tumour necrosis factor—TNF, interleukins—IL, etc.) implication in inflammation and degradative proteases [90]. 

Historically, with nutraceuticals being related to the natural components of the cartilage matrix (e.g., collagen, glucosamine and chondroitin), the study of their mechanism of action focused on structural (cartilage) effects [87].

Glucosamine and chondroitin are often used in combination. The primary interest of these products in osteoarthritic pain is their supposed anti-inflammatory properties. In fact, many in vitro and preclinical studies have shown their interaction in the nuclear factor-kappa B and p38 mitogen-activated protein kinase inflammatory pathways, as well as their involvement in the regulation of pro- and anti -inflammatory cytokines [91,92,93]. Glucosamine and chondroitin tend to stimulate, in in vitro and in vivo tests (mice and rat models), the expression of anti-inflammatory interleukins (IL-2, IL-10), reduce that of pro-inflammatory molecules (IL-1B, IL-6, TNF-α) and downregulate the production and expression of prostaglandin E_2_ synthetase and inducible cyclooxygenase (COX-2) or nitric oxide synthase (iNOS) [91,92,94,95,96]. Some antioxidant claims have also been made following in vitro results [97,98]. Finally, glucosamine and chondroitin are believed to modulate the expression and activity of certain catabolic enzymes implicated in the OA pathology. The results of different in vitro studies revealed, indeed, a decrease in the transcription and expression of degradative enzymes such as aggrecanases and matrix metalloproteinases (MMP-3, MMP-13) [92,94,99,100].

Collagen, especially in its hydrolysate form, will have the ability to prevent the destruction of cartilage through the production of macromolecules and suppression of catabolic enzymes. Many in vitro and preclinical studies have pointed to increased collagen type II synthesis, an important matrix component [101,102,103] and anti-inflammatory effects [104]. Another role, specific to collagen, is the oral tolerance phenomenon. It involves the intervention of the immune system and regulatory T cells (Tregs). The Tregs get activated by the collagen and are suspected to secrete many anti-inflammatory mediators upon meeting an articular cartilage (IL-4, IL-10, transforming growth factor-β) [105,106]. This modulation of the natural immune reaction is an important support for the anti-inflammatory activity and provides an environment conducive to cartilage repair. Moreover, it could be involved in the occurrence of adverse effects, which was elevated in human studies [107].

Omega-3s have evident anti-inflammatory properties through the reduction of IL-1α, IL-1β and TNF-α levels and the release of anti-inflammatory molecules [108,109]. In fact, the production of endogenous special proresolving mediators (SPMs) derived from these fatty acids helps to ease the inflammatory response in part responsible for osteoarthritic pain with even long-lasting effect shown [109,110]. Due to the competition for enzymes between omega-3 and omega-6 fatty acids, it has been suggested to promote the intake of high n-3/n-6 ratio diets to support the production of anti-inflammatory molecules and minimise the conversion of omega-6 in prostaglandins, leukotrienes and other pro-inflammatory lipoxygenase or COX by-products [111]. Omega-3s also seem to have anti-catabolic effects. Indeed, through in vitro and in vivo preclinical studies, the expressions of catabolic enzymes such as MMP-3, MMP-13 and ADAMTS-4/5 (a disintegrin and metalloproteinase with thrombospondin motifs) were downregulated [108,112,113]. More recently, the transient receptor potential vanilloid 1 (TRPV1) and the modulation of glial cells, both involved in pathologic pain, have been linked as a new target of omega-3 [114,115,116].

Many other nutraceuticals presented in vitro demonstration of anti-inflammatory, anti-oxidative and anti-catabolic properties. This was the case of hydroxycitric acid (extract from *Garcinia indica*) and other phytochemicals (*Boswellia serrata*, *Harpagophytum procumbens*, *Ribes nigrum*, *Salix alba*, *Brachystemma calycinum*, etc.). In recent years, the focus was on their anti-nociceptive properties, such as with cannabidiol. Cannabinoid pain-relieving effects are linked to various interactions and modulation of the endocannabinoid, inflammatory and nociceptive systems [117], with cannabidiol presenting high affinity for cannabinoid CB1 and CB2 receptors (antagonist), G-protein-coupled receptor 55 (antagonist) and many TRPV receptors (agonist) as well as peroxisome proliferator-activated receptor gamma [118], the latter two being largely recognised for their role in chronic pain and OA-related joint degradation [119]. Cannabinoids have shown great promises in animal models with acute and chronic pain [120,121,122].

Despite the importance of the data obtained and accumulated on these different mechanisms over the years, the application of this information remains limited. Most of the studies conducted on the mechanisms of action critically lacked about model and essay validity for the targeted OA pathology, as well as pharmacokinetics assessments. Several models of acute (inflammatory) pain have been used to represent OA, although this disease is much more complex, entangling chronic and degenerative conditions of many components, not just chondrocytes (in cell culture) [123]. These in vitro studies should be only kept producing precise mechanistic evidence of a chemical entity and then transposed into more complete models [124]. It has also been suggested that the use of models with naturally occurring disease provides the most valid models [125]. As for pharmacokinetics, the evaluation of the degree of systemic absorption and organ distribution is particularly lacking in nutraceuticals research. In fact, many products show very little systemic absorption, which inevitably translates to low efficacy. Is this the case of oral products based on glucosamine and chondroitin that showed relatively poor bioavailability in dogs (approximately 12% and 5% after a single dosing, respectively) [126]? Collagen-based products in rats, on the other hand, presented an absolute bioavailability of 58%, which is quite good [127].

### 4.12. General Discussions and Conclusions

Overall, previous systematic reviews support our findings [12,13,14]. However, a major difference between our systematic review and the previous ones is the number of articles identified. The most recent of these publications [14] was already from 2012; only 16 total publications were included, and their conclusions were based on the analysis of only one to four trials per nutraceutical. Surprisingly, 31 (out of 57) of the articles in our systematic review are dated from 2012 to the present, which is not consistent with previous review searches.

The quality of the studies we identified is often impoverished by the use of subjective and/or non-validated measurement tools. These tools, often carried out by owners who are not trained to complete them, are too susceptible to experimental bias and are not recommended in pain assessment according to recent professional guidelines [128,129]. In our assessments, we predetermined the degrees of reliability of the measurement tools. We prioritised objective pain quantification with kinetic or actimetric assessment methods, as these results are recognised as more valid and reliable (reference standard). Conversely, the subjective quantification of pain, very often estimated by the owner or veterinarian, shows less valid results and is more sensitive to the placebo effect [130] than objective methods [81].

Finally, several variables can influence the efficacy of nutraceuticals and thus affect the data that were evaluated. Among all the studies, we observed a wide variety of formulations (capsules, powdered food supplements, therapeutic diets, etc.). The mode of administration of nutraceuticals may affect the bioavailability of the nutraceuticals in the system and thus affect the physiological response observed [131]. Dosage, frequency and duration are also factors influencing the receipt of treatment. The studies analysed in this review had very variable treatment durations, ranging from about 1 to 6 months. As OA is a progressive disease, the duration of treatment is a key factor in the observation of pain clinical signs in pets [132]. Some lower-dose trials have shown a lack of effect probably due to dosage [25,56]. However, our results on the analysis of duration and dose on efficacy could not confirm the influence of these factors. This is probably due to the lack of power of analysis, related to the small sample size and huge variability in each category.

In addition, although some trials provide the same feeding bases as others, the content of each ingredient remains variable between studies and trials. For example, for two trials based on omega-3 polyunsaturated fatty acid supplementation, the content of eicosapentaenoic and docosahexaenoic acids may vary [133]. This example highlights the need for requirements on origin, standardised extraction and preparation methods. The content of the active ingredient and synergistic effects with other components of the formulation can also be a source of variability in expected results. 

The studies in this systematic review and meta-analysis greatly vary in their methodology. The development of clear norms and requirements that establish a standardisation of future clinical studies [85,134,135] will increase the quality and strength of evidence of efficacy and seek consensus on the true benefits of different nutraceuticals.

## 5. Conclusions

Our rigorous approach to meta-analysis allowed us to conclude with certainty that the use of omega-3 products beneficially modulates the painful condition of OA dogs and cats, while the intake of chondroitin-glucosamine has no analgesic effect. Further studies will be necessary to be able to state on the potential effects of collagen, cannabidiol and composite nutraceuticals, but these products seem promising.

## Figures and Tables

**Figure 1 ijms-23-10384-f001:**
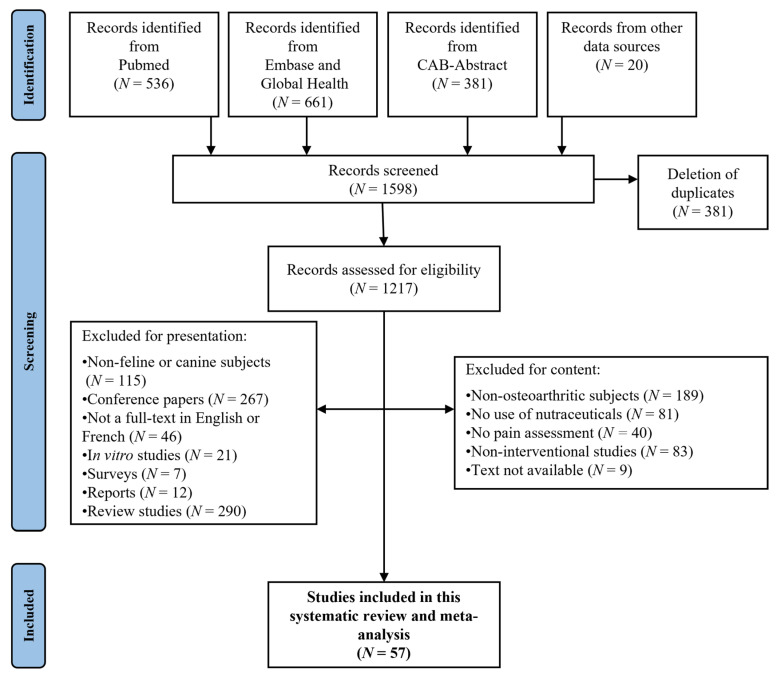
PRISMA flowchart of publications on use of nutraceuticals and therapeutic diets in canine and feline OA.

**Figure 2 ijms-23-10384-f002:**
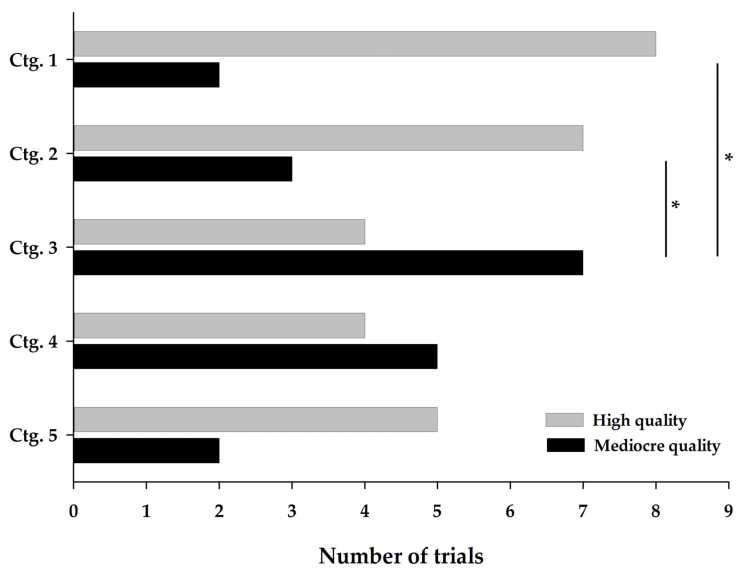
Distribution of quality levels of compound categories 1 to 5. A high-quality trial is represented by levels A and B, while a mediocre-quality trial is represented by levels C and D. Ctg. 1 (omega-3-enriched therapeutic diets), ctg. 2 (omega-3-based nutraceuticals), ctg. 3 (collagen-based nutraceuticals), ctg. 4 (chondroitin-glucosamine-based nutraceuticals) and ctg. 5 (cannabinoid-based nutraceuticals). Ctg., category. * indicates a significant difference (*p* < 0.05) between categories.

**Figure 3 ijms-23-10384-f003:**
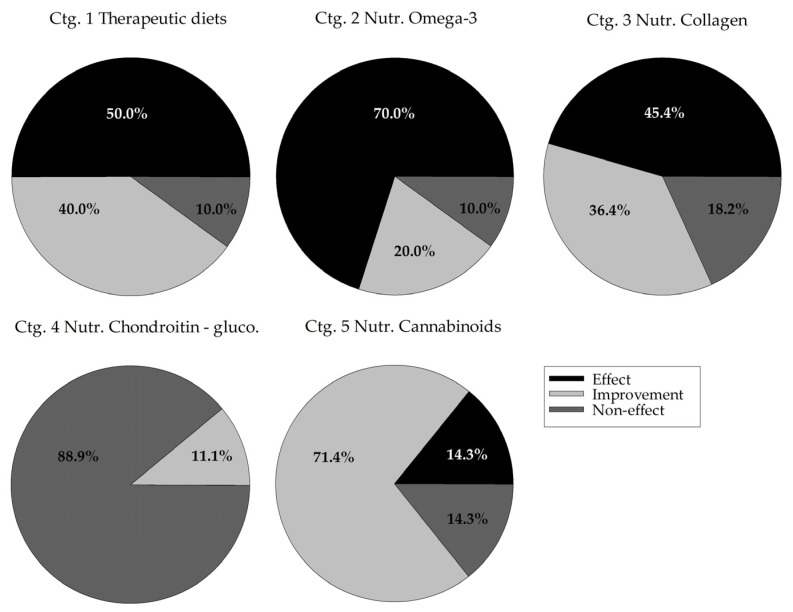
Distribution of efficacy levels of compound categories 1 to 5. On the total number of trials included in each category, expressed are the percentages of trials classified as an analgesic effect (vs. a control group), an improvement over time of the treatment group or a non-effect of the treatment. Ctg. 1 (omega-3-enriched therapeutic diets), ctg. 2 (omega-3-based nutraceuticals), ctg. 3 (collagen-based nutraceuticals), ctg. 4 (chondroitin-glucosamine-based nutraceuticals) and ctg. 5 (cannabinoid-based nutraceuticals). Ctg., category.

**Figure 4 ijms-23-10384-f004:**
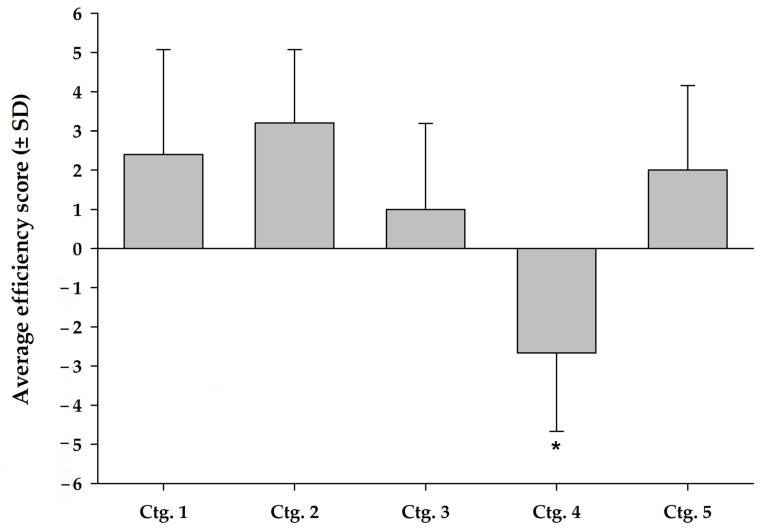
Average score (with standard deviation) of efficacy weighed for the quality level of categories 1 to 5. Weighted average efficacy score is plotted along with standard deviation of the score for each category. Ctg. 1 (omega-3-enriched therapeutic diets), ctg. 2 (omega-3-based nutraceuticals), ctg. 3 (collagen-based nutraceuticals), ctg. 4 (chondroitin-glucosamine-based nutraceuticals) and ctg. 5 (cannabinoid-based nutraceuticals). Ctg., category. * indicates significant difference (*p* < 0.001) vs. other categories.

**Table 1 ijms-23-10384-t001:** Quality assessment scale.

Criterion	Sub-Criteria (Score)
Risk of bias	1. Randomisation: Non-randomised (0), Not mentioned (0) or Randomised (2)
	2. Type of study: Single cohort (0), Cross-over (1) or Parallel (2) 3. Controlled study: No control group (0), Positive control (1*) or Placebo (1*) 4. Blinding procedure: Non-blinded (0), Single-blinded (1) or Double-blinded (2)
Methodological quality	5. Inclusion criteria: None (0), Other (1*), Experimental induction of OA in healthy animals (2), Owner-reported lameness (2*), Veterinary orthopaedic examination (2*), Inclusion grid (2*) or X-rays (2*) 6. Non-inclusion criteria: None (0), Weaning period too short (1*), Adequate weaning period (2*) or Description of non-inclusion criteria (2*)
	7. Exclusion criteria: None (0) or Description of exclusion criteria (2) 8. Control of possible bias: Non-randomised, or non-blinded, study with subjective assessment (0), Non-randomised, or non-blinded, study with objective assessments (1*), Research hypotheses and objectives clearly stated (0.5/each*), Ethics committee approval indicated (1*), Manuscript edited according to ARRIVE or CONSORT criteria (1*), Declaration of any conflict of interest (1*), Randomised, blinded study (2*) or No indication of the dose used (−5*) 9. Data collection and analysis: No information (0), Electronic collection, or methods already used (1), Quality assurance control (2*), Statistical analyses clearly described (1 or 2*)
Strengths of the scientific evidence	10. Sample size: <10 per group (0), Between 10 and 20 per group (2) or >20 per group (4) 11. Nature of data: Non-validated subjective (0*), Validated subjective (2*), Non-validated objective (1*) or Validated objective (4*) outcomes 12. Repetition of results obtained (according to the level of risk of bias): Only one study carried out (except if [A]) (0), Several studies [C] or [D] (1), One study [A] (2), Several studies [B or less] (3), Several studies [A and/or less] (4) or Several studies of level [A] (6)

[A] = Prospective, randomised, controlled, blinded study; [B] = prospective, randomised, observational cohort; [C] = non-randomised, controlled interventional trial (historical or prospective); [D] = cross-sectional study, or clinical case, or interventional trial, non-randomised, non-controlled. Scores followed by an asterisk (*) are cumulative and were therefore not exclusive.

**Table 3 ijms-23-10384-t003:** Weighing of efficacy scores in function of quality of each trial.

Quality of Trial	Level	Effect	Improvement	Non-Effect
Very high	A	+5	+3	−5
Good	B	+4	+2	−4
Medium	C	+2	+1	−2
Low	D	+1	+1	−1

## Data Availability

Not applicable.

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
