# Peer review of "A 2022 Systematic Review and Meta-Analysis of Enriched Therapeutic Diets and Nutraceuticals in Canine and Feline Osteoarthritis"

_ijms, 2022, doi:10.3390/ijms231810384_

Round 1
Reviewer 1 Report
The manuscript has merits for publication, pending some minor adjustments as indicated below:
Abstract: an introductory sentence should be added in order to contextualize the reader.
Line 22: Replace “the last” with “the latter.”
Introduction: More background information on the use of nutraceuticals to treat canine/feline OA should be added to highlight the importance of this systematic review.
Lines 37-41: Reference?
Line 60: Replace “affected with” with “affected by.”
Item 2.1: I’m surprised that terms like “bioactives,” “phytochemicals,” and other similar terms were not included in the search, as these compounds are the basis of nutraceutical development in several cases.
Lines 396-401: It is still unclear to me why omega-3 has this therapeutic effect. I believe the authors should expand a bit further on that.
Author Response
Please, see attached document

Reviewer 2 Report
In the present manuscript, Barbeau-Grégoire and co-workers, highlighted through a systematic review and meta-analysis studies that omega-3 enriched diets, omega-3 supplements and cannabidiol are the most effective analgesics in canine osteoarthritis pain management.
The article is well written and explained. I just have a recommendation:
Ø To complete the manuscript with information regarding the ability of omega-3 to modulate the intracellular signaling pathways associated with the pain control compare to collagen or chondroitin-glucosamine nutraceuticals.
Author Response
Please, see attached document
